# Comparative Meta-Analysis of Triplet vs. Quadruplet Induction Regimens in Newly Diagnosed, Treatment Naïve, Multiple Myeloma

**DOI:** 10.3390/cancers16172938

**Published:** 2024-08-23

**Authors:** Barry Paul, Faiz Anwer, Shahzad Raza, Aytaj Mammadzadeh, Bayan Khasawneh, Sara Shatnawi, Joseph McGuirk, Nausheen Ahmed, Zahra Mahmoudjafari, Muhammad Mushtaq, Al-Ola Abdallah, Shebli Atrash

**Affiliations:** 1US Myeloma Innovations Research Collaborative (USMIRC), Westwood, KS 66205, USA; anwerf@ccf.org (F.A.); razas@ccf.org (S.R.); nahmed5@kumc.edu (N.A.); zmahmoudjafari@kumc.edu (Z.M.); mmushtaq@kumc.edu (M.M.); aabdallah@kumc.edu (A.-O.A.); 2Department of Hematologic Oncology and Blood Disorders, Levine Cancer Institute, Atrium Health Wake Forest University School of Medicine, Charlotte, NC 28204, USA; 3Department of Hematology and Medical Oncology, Taussig Cancer Institute, Cleveland Clinic, Cleveland, OH 44195, USA; 4Division of Hematology and Oncology, Mayo Clinic, Rochester, MN 44905, USA; 5Department of Internal Medicine, Jordan University of Science & Technology, Irbid 22110, Jordan; 6Division of Hematologic Malignancies & Cellular Therapeutics, University of Kansas Medical Center, Westwood, KS 66205, USA; jmcguirk@kumc.edu

**Keywords:** multiple myeloma, anti-CD38 antibody, daratumumab, isatuximab

## Abstract

**Simple Summary:**

Induction regimens using 4-drugs for patients with newly diagnosed multiple myeloma have shown impressive results in multiple trials. However, trials of 4-drug regimens typically use a 3-drug regimen as their comparator, which limits our ability to determine whether the benefit is from the specific combination of drugs or just the addition of more agents. We performed a meta-analysis to compare all trials that compared 4-drug to 3-drug regimens and found that the addition of an anti-CD38 antibody (daratumumab or isatuximab) resulted in improved responses, while 4-drug regimens without an anti-CD38 antibody performed similarly to 3-drug regimens. While the addition of an anti-CD38 antibody led to increased adverse effects, these were predominantly mild and easily managed. These data suggest that an anti-CD38 antibody should be part of all 4-drug induction regimens for newly diagnosed myeloma patients.

**Abstract:**

The use of 4-drug induction regimens for treatment naïve newly diagnosed multiple myeloma (NDMM) is associated with improved depth of response and progression-free survival (PFS). However, head-to-head trials of 4-drug combinations are lacking, and instead, these regimens are typically compared to 3-drug backbones; limiting the ability to discern whether any additional benefit (or toxicity) is simply additive or represents a synergy (or interaction). We conducted a meta-analysis of phase 2 and phase 3 clinical trials that randomized treatment naïve NDMM patients to either a 4-drug or 3-drug induction regimen. We included 11 trials which represented 6509 unique patients. PFS for all trials in the meta-analysis was 54 months with a 4-drug induction and 8.9 months with a 3-drug induction (HR: 0.49; 95% CI: 0.45; 0.54), but there was no benefit to using a 4-drug induction that did not include an anti-CD38 antibody (PFS 4-drug 8.1 months, PFS 3-drug 8.0 months; HR 0.95; 95% CI 0.86; 1.06). Adverse events were more frequent with the quadruplet regimens but were predominately mild. High-grade (≥3) adverse events (AEs) that were more common with 4-drug regimens were infections (RR: 1.34; 95% CI 1.17; 1.54) and thrombocytopenia (RR: 1.39; 95% CI 1.12; 1.74). This study suggests that 4-drug induction regimens which include an anti-CD38 antibody improve efficacy although with additional toxicity in NDMM patients.

## 1. Introduction

Multiple myeloma (MM) is a malignancy of plasma cells that is characterized by clonal proliferation of terminally differentiated plasma cells within the bone marrow. For 2024, an estimated 35,780 new cases will be diagnosed in the United States, which represents 1.8% of all cancers and 19% of all hematologic malignancies [1]. With recent advances in therapy, patients can achieve long-term remissions, but eventually, relapses will occur. Still, the survival of MM patients has steadily increased over the past decade owing to novel combinations and newer classes of therapies during this time [2,3,4]. While the goal for any oncology treatment is an overall survival (OS) benefit, given the long remissions many NDMM patients achieve with induction therapy with or without autologous stem cell rescue, the continued advent of novel therapies available for relapsed refractory myeloma (RRMM), and high rates of crossover among treatment arms in clinical trials for MM patients, OS benefits are not commonly seen in trials of induction therapy for MM [5,6,7]. Additionally, it typically takes several years for OS data to mature, which may deprive patients from lifesaving novel therapies, particularly in areas of significant unmet medical need. Because of these challenges, randomized controlled trials for MM rarely incorporate OS as their primary endpoint [8]. Time to progression (TTP), progression-free survival (PFS), and more recently, minimal residual disease (MRD) negativity are commonly used as surrogate endpoints and have been validated in numerous retrospective analyses [9,10,11,12,13]. Still, an overall survival advantage should remain the gold standard for response assessment if feasible. Recently, the addition of the anti-CD38 antibody, daratumumab, to a triplet backbone (either lenalidomide, bortezomib, and dexamethasone [VRd] or thalidomide, bortezomib, and dexamethasone [VTd]) has shown an improved depth of response and PFS but has not shown an OS benefit as of yet [14,15]. However, trials comparing 4-drug regimens to each other are lacking, making it challenging to objectively assess what, if any, additional benefit adding more agents to standard-of-care therapies provides. In this meta-analysis, we examine the benefit of quadruplet vs. triplet induction to determine the PFS and OS benefit, if any, of a 4-drug combination. We also evaluate the toxicity profile of these regimens; recognizing that the addition of more agents to induction regimens may result in more adverse effects. 

## 2. Materials and Methods

Using the PubMed and Scopus databases, two researchers independently conducted a comprehensive systematic literature search and extracted the following information: title, first author’s name, treatment method, and outcome indicators. Inclusion criteria: We included prospective, randomized trials in transplant eligible NDMM patients, comparing 3-drug vs. 4-drug regimens with data on primary outcome of at least 2 years PFS. Secondary outcomes included data on MRD, OS and toxicity. The search encompassed studies treating multiple myeloma from 1 January 2010 to 31 December 2023. The following search terms: “(“Multiple Myeloma” [Mesh]) AND “Randomized Controlled Trial” [Publication Type].” and “Multiple myeloma” + “Randomized.” “Newly diagnosed multiple myeloma” was added to the Scopus search. Hence, the search resulted in fewer but focused publications for Scopus. Two researchers independently reviewed the publication lists for duplicates and only included studies comparing quadruplet regimens vs. triplets from English publications. This review was performed in accordance with the PRISMA (Preferred Reporting Items for Systematic Reviews and Meta-Analyses) guidelines and has not been registered.

The results of randomized controlled trials on time-to-event outcomes usually report the median time-to-event and Cox hazard ratio. The meta-analysis requires individual patient time-to-event data from the published Kaplan–Meier (KM) survival curves. Therefore, we used the modified iterative algorithm based on the Kaplan–Meier estimation method (modified-iKM) to obtain the survival coordinates as described previously [16]. Using a two-stage process, we determined iKM individual patient data (IPD). Stage 1 extracted quality data coordinates (time, survival probability) from published KM utilizing the DigitizeIt software package version 2.5 (http://www.digitizeit.de/, accessed on 1 March 2024), which analyzes the KM image from the final publication of each trial and provides coordinates for time and survival. In Stage 2, we processed these data coordinates using R software version 4.4.1 (https://www.R-project.org/, accessed on 1 March 2024), and the “IPDfromKM” package version 0.1.10 (https://CRAN.Rproject.org/package=IPDfromKM, accessed on 1 March 2024), then used the R “survival” package version 3.7-0 (https://CRAN.R-project.org/package=survival, accessed on 1 March 2024), to reconstruct the modified iKM. 

Finally, we performed a meta-analysis on the combined IPD using the R “meta” package version 6.5-0 (https://cran.r-project.org/web/packages/meta, accessed on 1 March 2024). The studies included were used to generate a forest plot to depict the meta-analysis results visually. A funnel plot was employed to evaluate potential publication bias, while sensitivity analysis assessed the stability of the findings. A *p*-value of <0.05 was used to determine statistical significance. Odds ratio (OR) and 95% confidence interval (CI) were employed for counting data to gauge the effect magnitude. Heterogeneity was quantitatively assessed using I2, categorizing low, medium, and high heterogeneities at 25%, 50%, and 75%, respectively [17]. An I2 < 50% indicated no significant heterogeneity, warranting using a fixed-effects model in the meta-analysis; otherwise, the study reported random-effects model results.

## 3. Results

### 3.1. Study Selection

Our electronic search yielded 1475 publications. Of these, 1372 were excluded as they did not report outcomes of quadruplet regimens. The remaining 103 trials were independently reviewed in full text by two investigators. One trial (IsKia) was excluded as it had not reported PFS data at the time of review [18]. An additional trial (GIMEMA-MM-03-05) only used maintenance for the 4-drug arm (and not the 3-drug arm), so it was excluded from the PFS analysis but was included in the toxicity and depth of response (defined as best overall response per IMWG 2016 criteria) assessments [19]. Another trial that investigated cyclophosphamide, bortezomib, and dexamethasone with or without clarithromycin was removed because it was stopped prematurely because of a lack of efficacy and increased toxicity with the 4-drug combination [20]. This resulted in 11 trials included in the toxicity analysis and 10 in the survival analysis (Figure 1 and Table 1). By pooling these studies, our toxicity analysis includes 6509 independent patients (3281 treated with a 3-drug regimen and 3228 treated with a 4-drug regimen); while our survival analysis includes 5330 independent patients (2699 treated with a 3-drug regimen and 2631 treated with a 4-drug regimen). All included studies were randomized open-label studies with no reported blinding of outcome assessment. While this may lead to an increased risk of a detection bias, given the heterogeneity of the included trials (including at least one where the 4-drug combination was the control arm), the risk of bias is thought to be minimal.

### 3.2. Meta-Analysis for Efficacy of 4-Drug vs. 3-Drug Induction Regimen in NDMM

Overall response rate (ORR), defined as a partial response (PR) or better, was similar between patients receiving a 4-drug vs. 3-drug induction regimen (RR: 1.09; 95% CI: 0.96; 1.24) when analyzed with a random effects model (Figure 2A). This was consistent with the results of the individual trials included in the meta-analysis, as only three trials (ALCYONE, Greek Myeloma Group, and EVOLUTION) showed a statistically significant benefit for a quadruplet induction, while the lone trial to show a significant benefit for a 3-drug induction was the MYELOMA IX trial. Meta-analysis of the four trials that reported ORR and included a 4-drug regimen that contained an anti-CD38 antibody compared to a 3-drug regimen without an anti-CD38 antibody showed no improvement in ORR with the 4-drug regimen (RR: 1.09; 95% CI: 1.00; 1.18). Similarly, there was no significant trend towards deeper responses (defined as ≥VGPR) using a random effects model in our meta-analysis (RR: 1.19; 95% CI: 1.00; 1.41) across all trials. Individually, eight of the eleven included trials showed a statistically significant improvement in the rate of ≥VGPR for the 4-drug regimen, while one study (the MYELOMA IX trial) showed the rate of ≥VGPR was higher with the 3-drug regimen (Figure 2B)**.** However, performing the same analysis only on the five trials that used a quadruplet induction containing an anti-CD38 antibody compared to 3-drug induction (not containing an anti-CD38 antibody) showed a statistically significant improvement in the depth of response with the addition of an anti-CD38 antibody (RR: 1.16; 95% CI: 1.04; 1.30). Conversely, an analysis of the six trials that did not include anti-CD38 antibody in either the 3-drug or 4-drug induction did not show an improvement with quadruplet induction (RR: 1.24; 95% CI: 0.87; 1.75).

A meta-analysis using a random effects model for MRD negativity (at 10^−5^) included the seven trials where this was reported noted significant improvement with a 4-drug induction regimen (RR: 1.43; 95% CI: 1.43–2.67) (Figure 2C). However, subgroup analysis showed that this effect was only observed in the five trials where an anti-CD38 antibody was included with the 4-drug induction (RR: 1.83; 95% CI: 1.33; 2.52) and was not seen in the two trials that used non-anti-CD38 antibody induction in either arm (RR: 2.30; 95% CI: 0.83; 6.38). 

A meta-analysis for PFS using a 4-drug vs. 3-drug induction showed significant improvement in the PFS with the 4-drug regimen (Figure 3A). Median PFS, using all trials, was 54 months with a 4-drug induction and 8.9 months with a 3-drug induction (HR: 0.49; 95% CI: 0.45; 0.54). When only including the trials that contained an anti-CD38 antibody in the 4-drug arm (Figure 3B), the median PFS was not reached, while it was 9.92 months in the triplet arm (HR 0.23; 95% CI: 0.19; 0.27). A meta-analysis of trials that did not include an anti-CD38 antibody in either arm (Figure 3C) showed no significant change in PFS with 4-drug vs. 3-drug induction (PFS 4-drug 8.1 months, PFS 3-drug 8.0 months; HR 0.95; 95% CI 0.86; 1.06).

### 3.3. Meta-Analysis for Toxicity of a 4-Drug vs. 3-Drug Induction Regimen in NDMM

Given the concern for additional toxicity with the addition of a fourth agent, we performed a meta-analysis to determine the increased risk for AEs with a 4-drug vs. 3-drug induction (Table 2). We found increased risk for all grades of infection (RR: 1.23; 95% CI 1.10; 1.38), diarrhea (RR: 1.12; 95% CI 1.02; 1.21), upper respiratory tract infections (RR: 1.60; 95% CI 1.31; 1.96), and thrombocytopenia (RR: 1.38; 95% CI 1.14; 1.66). Additionally, the use of a quadruplet induction was associated with increased rates of grade 3 or higher infections (RR: 1.34; 95% CI 1.17; 1.54) and thrombocytopenia (RR: 1.39; 95% CI 1.12; 1.74). No increased risk was observed for any or high grade: anemia, neuropathy, or pneumonia (Appendix A).

## 4. Discussion

Frontline therapy for NDMM has continued to evolve over the last 20 years. The development and widespread incorporation of newer classes of therapies, such as immunomodulatory agents, proteasome inhibitors, and anti-CD38 antibodies, into the management of NDMM has significantly improved outcomes in both transplant-eligible and ineligible patients. As newer therapies have become commercially available, these agents have often been added to standard-of-care backbones for NDMM in hopes of improved efficacy without significant additive toxicity. For instance, by the time thalidomide achieved its initial FDA approval in combination with dexamethasone for the treatment of NDMM in 2006 (becoming the first new drug approved for MM in over ten years), it had already shown positive results when combined with the then standard of care backbone of vincristine, doxorubicin, and dexamethasone (VAD) in a phase II trial, and was nearing the end of enrollment of a phase III randomized controlled trial of thalidomide plus VAD vs. VAD in NDMM [27,29]. Eventually, the FIRST trial established the superiority of lenalidomide and dexamethasone (Rd) over melphalan, prednisone, and thalidomide (MPT) [30]. Subsequently, the SWOG S0777 trial established the combination of lenalidomide, bortezomib, and dexamethasone (RVd) as the standard of care for NDMM based on the significant PFS and OS advantage compared to Rd, although with more adverse events seen in the triplet arm [31]. The S0777 trial enrolled patients between April 2008 and February 2012 and did not report out until 2016, underscoring the benefit of using MRD as a surrogate marker rather than waiting for PFS and OS data to mature, which is especially problematic in smaller phase II trials where limited events can have large implications on endpoints. Increasingly, data suggest that achieving a deeper remission to frontline therapy, as evidenced by the absence of MRD ideally at levels < 1 × 10^−6^, correlates with improved time to progression, PFS, and OS [10,32,33], suggesting that the goal of treatment in treatment naïve patients should be to induce the deepest remission possible. To that end, the use of 4-drug combinations as induction therapy for transplant-eligible NDMM has become more ubiquitous based on the results of multiple phase II and III clinical trials. Several other anti-myeloma therapies have similarly been added to standard-of-care doublets and triplet induction therapies in both transplant-eligible and transplant-ineligible NDMM [15,21,22,31,34,35,36,37]. Specifically, the addition of the anti-CD38 antibodies daratumumab and isatuximab to multiple triplet backbones has shown unrivaled depth of response and PFS advantage [15,18,23,28,35,36,37]. While these trials were largely positive, the additional toxicity, impaired quality of life (QoL), and costs associated with adding more agents must be considered [38,39]. One question that often arises when comparing trials with unbalanced arms (i.e., 3 agents vs. 2 or 4 agents vs. 3) is whether any additional benefit observed is additive or synergistic. Ideally, new combinations are nominated based on preclinical modeling and complementary mechanisms of action. However, even in this setting, more therapy does not always equate with improved efficacy. An example was the S1211 trial, which investigated the use of the anti-SLAMF7 antibody elotuzumab in combination with RVd (Elo-RVd) compared with RVd alone in high-risk NDMM and failed to show additional benefit with elotuzumab compared to the triplet in this setting [22]. Still, the overwhelming trend in current therapy is incorporation of additional drugs into standard of care regimens for NDMM. 

To address this challenge, our meta-analysis of multiple trials that randomized NDMM patients to either a 4-drug or 3-drug induction tried to answer the scientifically relevant question of whether a 4-drug regimen is more effective without inducing additional high-grade AEs. Additionally, we further conducted subgroup meta-analyses to determine which agent(s) were primarily responsible for any additional benefit or toxicity. Interestingly, our results suggest that the incremental benefit of the 4-drug regimen is mainly due to the addition of an anti-CD38 antibody. Using the same analysis parameters but only including the five trials that did not include an anti-CD38 antibody as part of the quadruplet regimen yielded nearly identical PFS to the pooled respective 3-drug regimens. Similarly, we did not observe a discernible benefit in ORR with a 4-drug regimen (regardless of whether an anti-CD38 antibody was included), but we observed a significant improvement in depth of response and achievement of higher MRD negativity rates in patients treated with a 4-drug induction only when the induction regimen contained an anti-CD38 antibody. This likely translated to PFS benefit observed. This is further supported by the unparalleled PFS advantage seen with the combination of daratumumab, lenalidomide, and dexamethasone (Dara-Rd) compared with Rd in transplant-ineligible NDMM patient (PFS of 61.9 months for Dara-Rd vs. 34.4 months for Rd) [40]. Surprisingly, our data also show discordance between the overall response rate and PFS as there was no improvement in the depth of response in our meta-analysis, even with an anti-CD38 containing regimen, while we detected a highly significant PFS advantage with the use of daratumumab. However, MRD negativity rates were significantly higher in the 4-drug regimens containing an anti-CD38 antibody. MRD negativity has been established as an important surrogate endpoint in multiple trials and is now guiding treatment decisions in some trials, so this discordance is surprising [41,42,43]. This discordance between ORR and MRD negativity in our meta-analysis samples is likely related to the heterogeneity among maintenance therapies used in the included trials and the somewhat limited sample size. Ultimately, overall survival (OS), when mature, will be the best differentiator of regimens. However, given the abundance of new and emerging therapies in MM, it is not surprising that a recent retrospective analysis demonstrated a poor correlation between PFS and OS in NDMM patients [44].

While AEs were more common with 4-drug regimens compared with 3-drug regimens, the differences were similar regardless of whether the 4-drug regimen included an anti-CD38 antibody. Yamamoto and colleagues recently performed a cost analysis of daratumumab as part of a Dara-RVd-based quadruplet induction for NDMM. Based on their assumptions of subsequent lines of therapy and duration of benefit, the cost of using RVd as induction would surpass that of Dara-RVd after 33 cycles. As the PFS of Dara-RVd in the GRIFFIN trial has far exceeded this timepoint (4-year PFS is 87% for Dara-RVd vs. 70% for RVd), the expected savings based on this model would be substantial [39]. These data strongly support using a 4-drug induction regimen containing an anti-CD38 antibody.

## 5. Limitations

This study has several important limitations. While combining data for several trials helps to limit bias that may be inherent in individual trials, there is still a risk that this could affect the outcomes of the meta-analysis. Additionally, several of the older trials included in the meta-analysis use what would be considered an inferior regimen to the more novel therapies commonly represented in the new trials. Still, by performing a meta-analysis, our hope is that this effect is diluted and still helps to support our conclusion that not all 4-drug regimens will outperform all 3-drug regimens, which needs to be considered when subjecting patients to additional costs and toxicity. Another noticeable limitation is the overall heterogeneity seen in trials. Specifically, there is significant variation in the cycle length, number of cycles, doses of immunomodulatory drugs and proteasome inhibitors, use of consolidation therapies, maintenance regimens, and variable outcomes between different trials. This is reflected by high I^2^ statistics. To make the comparison more accurate and decrease bias, we included only randomized trials in our analysis. Therefore, both arms were equally balanced based on the original randomization of the clinical trial.

## 6. Conclusions

The present study strongly supports the use of a 4-drug induction regimen containing an anti-CD38 antibody in treatment naïve NDMM patients, given the improvement in deeper responses and longer PFS and limited additional toxicity. In the absence of head-to-head trials comparing novel therapies, meta-analysis can be used to determine the rationale combinations of anti-myeloma therapy using the surrogate markers that have been presented. However, revisiting the overall survival signal is crucial once these data mature. 

## Figures and Tables

**Figure 1 cancers-16-02938-f001:**
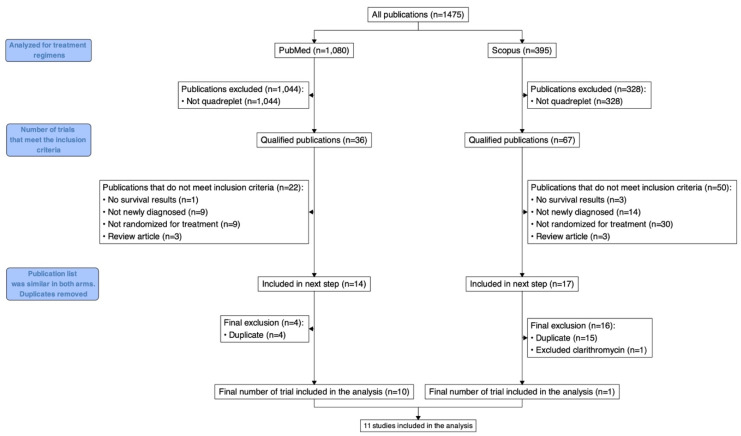
Preferred reporting items for systemic reviews and meta-analysis (PRISMA) statements.

**Figure 2 cancers-16-02938-f002:**
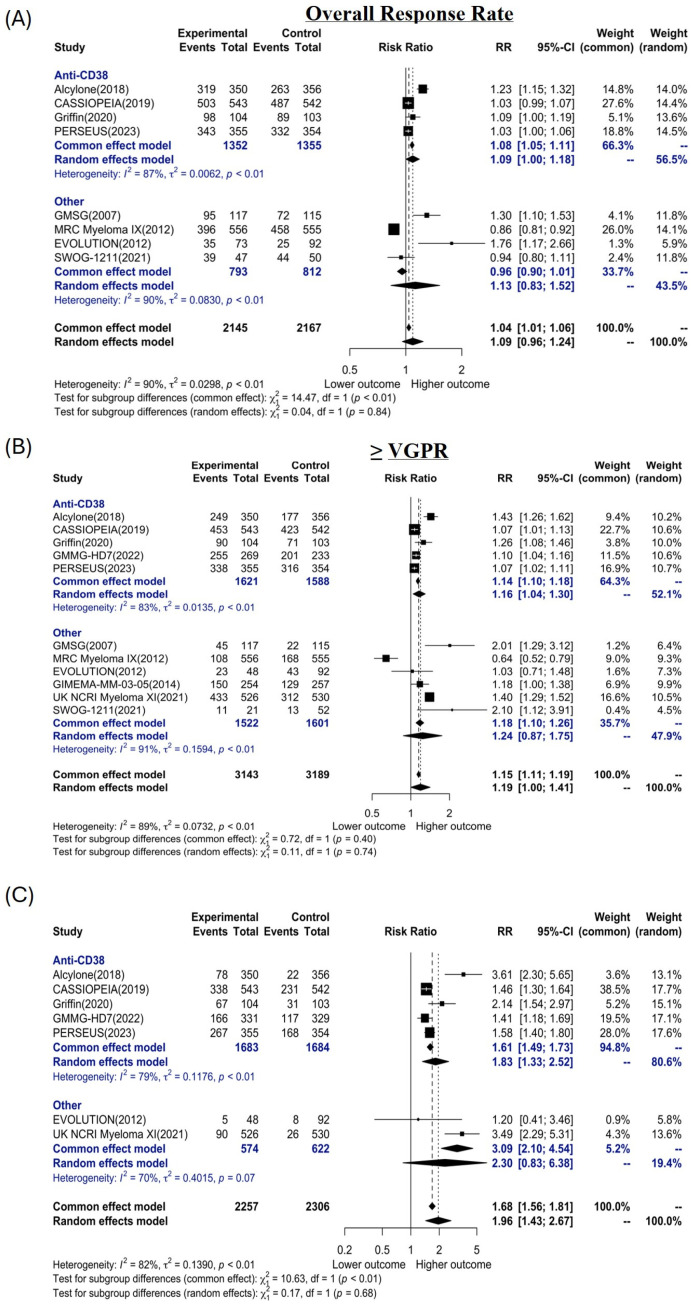
Forest plot of response overall response rate (**A**), ≥VGPR rate (**B**), and MRD negativity rate (**C**) in 4-drug vs. 3-drug induction regimens for newly diagnosed myeloma grouped by presence or absence of CD38 antibodies (Anti-CD38. Risk ratios in black represent values for individual trials whereas risk ratios in blue represent the common and random effects models for each subgroup. Values in bold represent risk ratios for common or random effects models across all trials-regardless of the presence or absence of an anti-CD38 antibody. See Table 1 for references.

**Figure 3 cancers-16-02938-f003:**
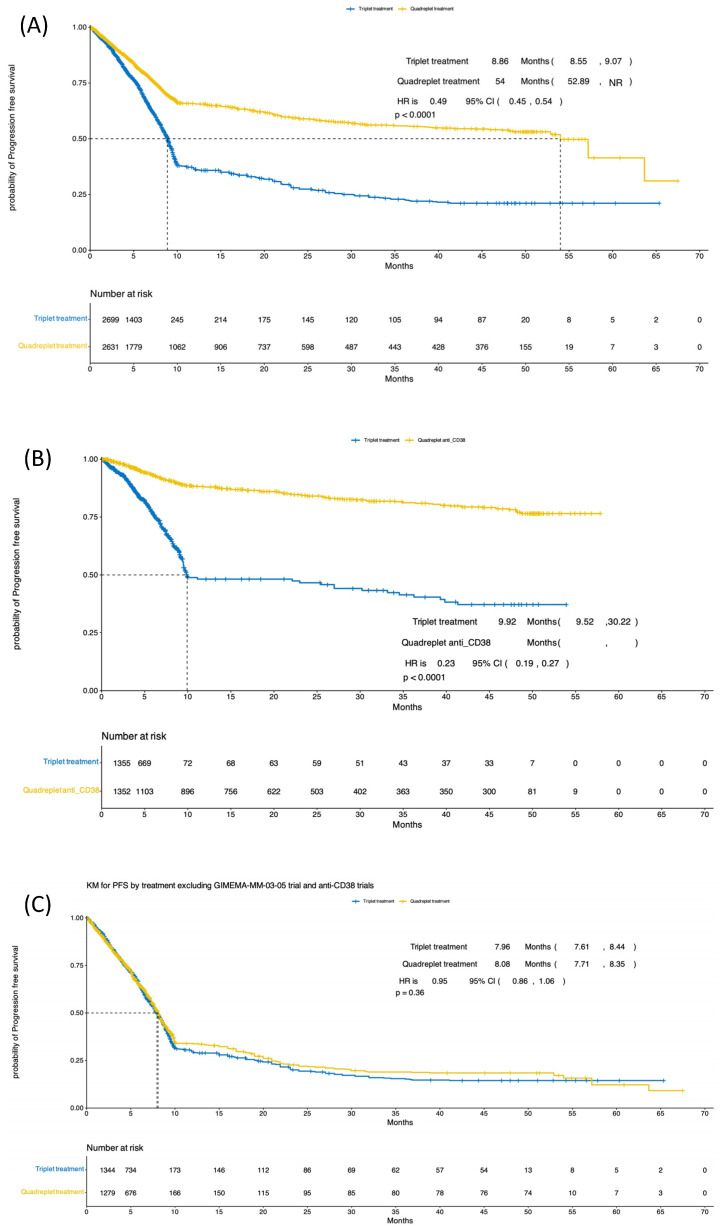
PFS in (**A**) all 4-drug vs. 3-drug trials included in the meta-analysis, (**B**) in trials where the 4-drug regimen included an anti-CD38 antibody, and (**C**) trials where the 4-drug combination did not include an anti-CD38 antibody (NR = not reached).

**Table 1 cancers-16-02938-t001:** Studies included in meta-analysis (see references for details).

Study	Phase	Population	3-Drug Arm (N)	4-Drug Arm (N)	MRD Reported?	Reference
Alcyone	3	NDMM	VMP (356)	Dara-VMP (350)	Yes	[21]
CASSIOPEIA	3	NDMM	VTd (542)	Dara-VTd (543)	Yes	[14]
S1211	2	HR-NDMM	RVd (52)	Elo-RVd (48)	No	[22]
PERSEUS	3	NDMM	RVd (354)	Dara-RVd (355)	Yes	[23]
GRIFFIN	2	NDMM	RVd (103)	Dara-RVd (104)	Yes	[15]
MYELOMA XI+	3	NDMM	RVd (265)TVd (265)	KRD-Cy (526)	Yes	[24]
MYELOMA IX	3	NDMM	Cy-Td (555)	Cy-VAD (556)	No	[25]
EVOLUTION	2	NDMM	RVd (42)Cy-Vd (50)	Cy-RVd (48)	Yes	[26]
GREEK MM STUDY GROUP	3	NDMM	VAd (115)	T-VAd (117)	No	[27]
GIMEMA-MM-03-05	3	NDMM	VMP (253)	T-VMP (250)	No	[19]
GMMG-HD7	3	NDMM	RVd (329)	Isa-RVd (331)	Yes	[28]

**Table 2 cancers-16-02938-t002:** Percent of all grade and grade ≥ 3 toxicities observed in both 3-drug and 4-drug induction regimens in trials included in the meta-analysis (NR = not reported).

	Alcyone		CASSIOPEIA		S1211		PERSEUS		GRIFFIN		MYELOMA XI+		MYELOMA IX		EVOLUTION		GREEK MM STUDY GROUP		GIMEMA-MM-03-05		GMMG-HD7	
	3-Drug (N = 354)	4-Drug (N = 346)	3-Drug (N = 542)	4-Drug (N = 543)	3-Drug (N = 52)	4-Drug (N = 48)	3-Drug (N = 354)	4-Drug (N = 355)	3-Drug (N = 103)	4-Drug (N = 104)	3-Drug (N = 530)	4-Drug (N = 526)	3-Drug (N = 555)	4-Drug (N = 556)	3-Drug (N = 92)	4-Drug (N = 48)	3-Drug (N = 115)	4-Drug (N = 117)	3-Drug (N = 253)	4-Drug (N = 250)	3-Drug (N = 329)	4-Drug (N = 331)
Neutropenia																						
All grades	52.5	49.7	16.4	28.9	32.7	37.5	57.6	68.5	35.0	54.8	45.1	48.7	NR	NR	NR	NR	24.3	17.9	NR	NR	7.0	23.3
Grades 3/4	38.7	39.9	14.6	27.3	9.6	16.7	50.0	61.4	21.4	39.4	17.2	16.0	NR	NR	19.6	43.8	10.4	7.7	28.1	38.4	7.0	23.3
Anemia																						
All grades	37.6	28.0	NR	NR	1.9	0.0	20.3	22.0	32.0	33.7	72.8	77.2	NR	NR	NR	NR	NR	NR	NR	NR	6.1	3.9
Grades 3/4	19.8	15.9	NR	NR	0.0	0.0	6.2	5.9	5.8	8.7	5.1	9.9	NR	NR	5.4	8.3	NR	NR	9.9	10.0	6.1	3.9
Thrombocytopenia																						
All grades	53.7	48.8	13.5	20.1	61.5	54.2	33.6	47.9	35.0	41.3	23.4	48.7	NR	NR	NR	NR	13.0	9.4	NR	NR	4.6	6.6
Grades 3/4	37.6	34.4	7.4	10.9	19.2	20.8	16.9	28.7	8.7	15.4	1.7	5.9	NR	NR	9.8	14.6	7.8	4.3	19.8	22.0	4.6	6.3
Neuropathy (%)																						
All grades	34.2	28.3	62.7	57.8	61.5	79.2	50.6	53.0	71.8	56.7	35.1	20.0	NR	NR	70.7	68.8	46.1	12.8	NR	NR	31.9	27.2
Grades 3/4	4.0	1.4	8.5	8.7	11.5	8.3	4.0	4.2	7.8	6.7	0.8	0.2	NR	NR	14.1	12.5	6.1	0.9	5.1	10.8	7.6	6.6
Infections																						
All grades	48.0	66.8	56.5	64.6	28.8	25.0	75.1	85.9	61.2	86.5	NR	NR	NR	NR	NR	NR	11.3	15.4	NR	NR	22.8	25.4
Grades 3/4	14.7	1.4	21.8	19.3	15.4	8.3	26.8	34.9	21.4	22.1	NR	NR	NR	NR	NR	NR	6.1	8.5	9.1	12.8	9.7	12.1
URI																						
All grades	13.8	26.3	NR	NR	7.7	6.3	24.6	31.3	43.7	59.6	NR	NR	NR	NR	NR	NR	NR	NR	NR	NR	NR	NR
Grades 3/4	1.4	2.0	NR	NR	0.0	0.0	1.7	0.6	1.9	1.0	NR	NR	NR	NR	NR	NR	NR	NR	NR	NR	NR	NR
Pneumonia																						
All grades	4.8	15.3	1.7	3.5	0.0	0.0	10.7	18.0	10.7	8.7	16.0	16.2	NR	NR	NR	NR	NR	NR	NR	NR	NR	NR
Grades 3/4	4.0	11.3	NR	NR	0.0	0.0	5.9	10.4	10.7	7.7	7.9	10.3	NR	NR	3.3	4.2	NR	NR	2.4	5.6	NR	NR
Diarrhea																						
All grades	24.6	23.7	NR	NR	73.1	85.4	53.1	60.3	49.5	56.7	23.2	28.7	NR	NR	NR	NR	NR	NR	NR	NR	NR	NR
Grades 3/4	3.1	2.6	NR	NR	3.8	14.6	7.6	0.8	3.9	49.0	1.7	2.9	NR	NR	3.3	6.3	NR	NR	2.8	1.6	NR	NR

## Data Availability

Data available on request.

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
