# Peer review of "Comparative Meta-Analysis of Triplet vs. Quadruplet Induction Regimens in Newly Diagnosed, Treatment Naïve, Multiple Myeloma"

_cancers, 2024, doi:10.3390/cancers16172938_

Round 1

Reviewer 1 Report

Comments and Suggestions for Authors

Very well written meta-analysis in the field of myeloma that can change the current practice guidelines and quality of care for patients.

Author Response

Comment 1: Very well written meta-analysis in the field of myeloma that can change the current practice guidelines and quality of care for patients.

Response 1: We thank the reviewer for their kind words and agree with the importance of the findings detailed in the manuscript.

Reviewer 2 Report

Comments and Suggestions for Authors

1 -  This is an excellent manuscript addressing the conclusion that the addition of CD-38 to a drug regimen for multiple myeloma, improved the effect of therapy.

2-  The authors clearly demonstrated  as seen in Figure 3. 

3-  Other Figures are also well constructed, ( Figure 2 is a little cumbersome, but it just takes a little time to understand it). 

4-  I would have liked to have seen a little more detail on the" mechanisms" of action of CD38 on efficacy of treatment, but perhaps this would be the subject of another manuscript.

5-  The references are are current, and provide excellent information.

Author Response

Comment 1: This is an excellent manuscript addressing the conclusion that the addition of CD-38 to a drug regimen for multiple myeloma, improved the effect of therapy.

Response 1: We thank the reviewer for their kind words and agree with their conclusion

Comment 2: The authors clearly demonstrated as seen in Figure 3.

Response 2: Thank you.

Comment 3: Other Figures are also well constructed, (Figure 2 is a little cumbersome, but it just takes a little time to understand it).

Response 3: Thank you for your comment. We agree that there is a lot of information presented in figure 2 and that the figure is a little “busy”, but overall we think it presents important information in the most clear and concise way.

Comment 4: I would have liked to have seen a little more detail on the "mechanisms" of action of CD38 on efficacy of treatment, but perhaps this would be the subject of another manuscript.

Response 4: Thank you for the suggestion. While the mechanisms of action of daratumumab are myriad and have been detailed in other publications the specific mechanism of synergy with the various combinations included in the meta-analysis are less clear and are unfortunately beyond the scope of our study.  We agree that this would be an excellent topic for a future manuscript and will investigate this opportunity in the future.

Comment 5:  The references are current and provide excellent information.

Response 5: Thank you.

Reviewer 3 Report

Comments and Suggestions for Authors

The study demonstrates the result of a meta-analysis of multiple myeloma.

1. In result section, the meaning of "depth of response assessments" is unclear in line 119. Some explanation of the response assessment is needed and how to measure the depth of the response assessments may be described around the sentence. 

2. Selected toxicities seen in included trials in Table 2 may be revised to indicate what the value indicates, and the size of the value signifies.

3. It is very interesting that increased risk for all grade of infection was observed, while the risk for Grades 3/4 of infection seems to be decreased in Alcyone, CASSIOPEIA and S1211. Some discussion may be added.

Author Response

Comment 1:  In result section, the meaning of "depth of response assessments" is unclear in line 119. Some explanation of the response assessment is needed and how to measure the depth of the response assessments may be described around the sentence

Response 1: Thank you for this comment. We have updated the manuscript to reflect that this represents the best response per IMWG 2016 criteria.  

Comment 2: Selected toxicities seen in included trials in Table 2 may be revised to indicate what the value indicates, and the size of the value signifies

Response 2: Thank you for pointing this out. We have revised the table legend to clarify that this table represents the percent of all grade and grade 3/4 toxicities reported in each trial included in the meta-analysis.

Comment 3: It is very interesting that increased risk for all grade of infection was observed, while the risk for Grades 3/4 of infection seems to be decreased in Alcyone, CASSIOPEIA and S1211. Some discussion may be added.

Response 3: This is a very interesting observation and underscores the value of meta-analysis. We feel that this likely represents differences in anti-infective prophylaxis, timing of trials (i.e.  before or during the SARS-CoV-2 pandemic) and potential for additive or synergistic toxicities with various combinations. As our study is purely a meta-analysis, we felt it was out of the scope of our manuscript to present these hypotheses.